# Physiological and Transcriptome Analyses of CaCl_2_ Treatment to Alleviate Chilling Injury in Pineapple

**DOI:** 10.3390/plants11172215

**Published:** 2022-08-26

**Authors:** Mengzhuo Zhang, Qiang Zhang, Cong Tian, Guangsen Liu, Yonggui Pan, Xiangbin Xu, Xuequn Shi, Zhengke Zhang, Lanhuan Meng

**Affiliations:** 1School of Food Science and Engineering, Hainan University, Haikou 570228, China; 2School of Biology and Basic Medicine Science, Soochow University, Suzhou 215006, China

**Keywords:** pineapple, CaCl_2_ treatment, chilling injury, ROS, oxidative and antioxidants, transcriptome analysis

## Abstract

The post-harvest ripening of pineapples can be effectively postponed by refrigerated storage. Nevertheless, internal browning (IB) frequently appears in pineapples after refrigerated storage during the course of the shelf life at room temperature, which is known as chilling injury (CI). In this study, the chilling injury of pineapple fruit was induced by a low temperature (6 °C) and transferred to normal-temperature storage; the best concentration of 50 μmol/L of CaCl_2_ was selected by the IB appearance and electrical conductivity. Fruit quality, reactive oxygen species (ROS), antioxidants, and transcription factors were investigated. The physiological data results indicated that pineapples treated with 50 μmol/L of CaCl_2_ maintained fruit quality, decreased reactive oxygen species (ROS), and enhanced the antioxidant activity of fruits, alleviating internal browning (IB) symptoms in pineapple fruit. The expressions of related genes were also consistent with the physiological changes by the transcriptome data analysis. In addition, we focused on some related metabolic pathways, including phenylpropanoid biosynthesis, MAPK pathway, plant hormone, plant–pathogen interaction, tricarboxylic acid cycle (TAC), and fatty acid biosynthesis. We performed integrative analyses of transcriptome data combined with a series of physiology and experimental analyses on the internal browning of pineapples, which will be of great significance to extending the shelf life of pineapples through molecular biology in the future.

## 1. Introduction

The pineapple (*Ananas comosus* L. Merr.) is one of three major tropical economic crops (among the most significant harvest crops) and a tasty tropical fruit [1]. To ensure long-distance transport trade, the harvested fruits are usually stored in low-temperature conditions. However, it is easy to induce the internal browning of pineapple fruit, which has a negative effect on the commodity value and the price of this fruit [2]. In addition to pineapple, these unfavorable changes (caused by cold storage) are common in many other harvested horticultural fruits, including apples, pears, peaches, and bananas [3,4,5,6],. Internal browning, also termed endogenous browning or black heart, is a physiological disorder of pineapple fruit [7]. To date, the mechanism of harvested fruit browning has not been elucidated under cold storage.

Cold storage for tropical fruits leads to a series of physiological disorders within plant cells, including dehydration, flesh browning, and so on [8,9]. These processes are also closely related to the compositions of various flavor-related metabolites and phytohormones, such as sugars, fatty acids (FAs), amino acids, ethylene (ET), jasmonate (JA), abscisic acid (ABA), salicylic acid (SA), and gibberellic acid (GA), which lead to flavor deterioration and cold-stress induced modulation in plants [3,10]. Several researchers have observed that flavor-related metabolite compositions (sugars) will stabilize the membrane system and influence commercial quality and shelf life [11]. Phytohormones, as chemical signal molecules, modulate pro-metabolism to increase stress tolerance in plants [12]. For example, exogenous ABA inhibits the internal browning of harvested pineapple fruit [13]. In addition, cold storage will decrease the quality of fruit, which is associated with the complex regulation of genes in multiple metabolism pathways, including those related to cell wall metabolism, plant hormone signal transduction, starch and sucrose metabolism, lipid metabolism, phenylpropanoid biosynthesis, and the MAPK signaling pathway [14,15,16,17,18]. Meanwhile, transcription factor genes (CRCK2, MYB, P450, AOS, WRKY, NAC) enhance cold resistance by inducing the expression of the downstream target genes [15].

Ca^2+^ is recognized as a crucial second messenger in signaling pathways and performs a regulating function in the cold stress response in plants [19]. Research indicates that deficiencies in calcium implicate physiological disorders in apples [20]. Calcium, in addition to its major role in cross-linking pectin chains in plant cell walls, also plays an important regulatory role in cell metabolism, such as in protein kinase signaling [21]. Wei and Zhao noted that exogenous calcium could alleviate cold injuries by reducing electrolyte leakage and preserving cell membrane integrity [22]. More extensive applications of calcium salt in the fruit have alleviated abiotic stress. However, this is only in regard to physiological and biochemical aspects; to date, the mechanism of internal browning during calcium treatment in post-harvest storage at low temperatures has not been elucidated at the molecular level.

In this study, an integrated analysis of the pineapple’s physiological biochemical and transcriptome factors was an effective way to understand the genome function and the related pathways. Specifically, IB was induced by incubating green stage pineapple fruit at a low temperature (6 °C) for 1 and 2 w followed by room temperature (25 °C) for an additional 3 and 6 d after postharvest. We compared the effects of postharvest treatments between CaCl_2_ solutions and the control group pineapple fruit regarding the incidences of IB following cold storage (6 °C) after 1 and 2 w and the transfer to room temperature (0, 3, and 6 d). We specifically analyzed the flesh quality physiological indices, such as ROS (H_2_O_2_, O_2_^• −^), membrane lipid peroxidation (MDA), antioxidant compound (ASA, GSH, TP), oxidase (PPO) antioxidant enzyme activities (POD, CAT, GR, SOD, APX); combinations with transcriptome data analyses explained the internal molecular mechanism of the fruit’s IB regulated by calcium ion. These findings provide new insight into the molecular mechanisms of chilling-induced pineapple IB.

## 2. Results

### 2.1. Physiological and Biochemical Results 

#### 2.1.1. IB Appearance and Electrical Conductivity

The IB appearance of pineapple fruit showed 1 w + 0 d and 2 w + 0 d of no browning symptoms; when transferred to room temperature, IB symptoms occurred. The longer the refrigeration, the more serious the chilling injury. From Figure 1A, the pineapple treated with 50 μmol/L of CaCl_2_ had a moderate chilling injury. 

As shown in Figure 1B, for the control and treatment groups, concerning relative conductivity, there was a continuous increase during refrigeration after 1 and 2 w; however, 50 μmol/L of CaCl_2_ is a lower electronic leakage rate. Respectively, the value of 42.08 ± 0.0047% reached 71.99 ± 0.0248% at 1 w and the value of 64.24 ± 0.0108% reached 77.05 ± 0.0125% at 6 d. The control group electronics leakage rate was higher at any storge time. Moreover, 50 μmol/L of CaCl_2_ can strikingly decrease the relative conductivity in pineapple fruit and reduce the membrane damage to fruit storage.

By the IB appearance and electrical conductivity, 50 μmol/L of CaCl_2_ is the best concentration selection. Therefore, the following physiological indicators and transcriptome data are only for the 50 μmol/L CaCl_2_-treated group and the control group.

#### 2.1.2. Membrane ROS Production and Membrane Lipid Peroxidation

As shown in (Figure 2A,B), the O_2_^• −^ production rate and H_2_O_2_ content in fruit exhibited overall increments in fruit DP during room temperature storage, but in the HP O_2_^• −^ production rate, there was no significant difference (Figure 2A). The H_2_O_2_ content of the control group was more than the 50 μmol/L CaCl_2_-treated group in the HP during storage. The DP control group was significantly greater than the 50 μmol/L CaCl_2_-treated group after 2 w from 0.57 ± 0.014 umol/g of FW to 0.73 ± 0.023 umol/g of FW. In conclusion, 50 μmol/L of CaCl_2_ reduced the ROS damage to fruit cells.

MDA content in both the control group and the 50 μmol/L of the CaCl_2_-treated group continuously increased, but DP had higher HP 3 and 6 d after the transfer from refrigeration to room temperature (Figure 2C). This proves that pineapple fruit is subjected to membrane lipid peroxidation during storage (from cold storage to room temperature).

#### 2.1.3. Ascorbic Acid (ASA) and Glutathione (GSH) Contents

Initially, the ascorbic acid content of the control and the 50 μmol/L of CaCl_2_-treated fruit were approximately 2.72 ± 0.160 g Kg^−1^ and 2.72 ± 0.071 g Kg^−1^ in refrigeration (Figure 3A). Following the storage time elongation at room temperature, the overall increments of the ASA content in HP. Moreover, the 50 μmol/L CaCl_2_-treated group was sharply higher than the control group. Comparatively, the DP was lower during most storage time periods. However, 50 μmol/L of the CaCl_2_ group was more than the control group.

In Figure 3B, regarding the GSH content, there are no significant changes in HP, but in DP, the 50 μmol/L CaCl_2_-treated group had a higher GSH content than the control group; the value was from 0.099 ± 0.0013 g Kg^−1^ to 0.013 ± 0.0010 g Kg^−1^ after 1 w and from 0.089 ± 0.0034 g Kg^−1^ to 0.082 ± 0.0096 g Kg^−1^ after 2 w. Based on the change trajectory, the control group’s HP decreased after 1 and 2 w, but the HP of the 50 μmol/L CaCl_2_-treated group experienced a slower change (a ‘rise again’ phase); this was, respectively, 1 w + 6 d and 2 w + 3 d. It shows that the 50 μmol/L CaCl_2_-treated group had more stress ability to cold stress.

#### 2.1.4. Phenolic Metabolism and Active Oxidase

As shown in Figure 4A, total phenolic content decreased after an initial increase in the HP of the control group and the 50 μmol/L CaCl_2_-treated group after 1 w. Moreover, the HP of the 50 μmol/L CaCl_2_-treated group experienced relative lagging. Respectively, the HP initial values of the control group and the 50 μmol/L CaCl_2_-treated group were 0.589 ± 0.0406 g Kg^−1^ and 0.525 ± 0.0277 g Kg^−1^ (1 w + 0 d), then rose to 0.956 ± 0.0225 g Kg^−1^ and 0.818 ± 0.0811 g Kg^−1^ (1 w + 3 d), and finally went down to 0.229 ± 0.0189 g Kg^−1^ and 0.551 ± 0.0132 g Kg^−1^ (1 w + 6 d). The total phenolic content continuously decreased in the control group HP after 2 w; the value went from 0.738 ± 0.0143 g Kg^−1^ to 0.216 ± 0.0100 g Kg^−1^. However, the HP of the 50 μmol/L CaCl_2_-treated group was first up and then down from 0.410 ± 0.0262 g Kg^−1^ (2 w + 0 d) to 0.836 ± 0.0151 g Kg^−1^ (2 w + 3 d) to 0.622 ± 0.0060 g Kg^−1^ (2 w + 6 d). Evidently, the HP of the 50 μmol/L CaCl_2_-treated group was higher than the HP of the control group (2 w + 3 d and 2 w + 6 d). There is no obvious difference in the fruit DP. 

As shown in Figure 4B,C, the PPO activity of the DP fruit was higher than the HP fruit. For HP fruit, there was no difference for both the control group and the 50 μmol/L CaCl_2_-treated group in 1w; however, the 50 μmol/L CaCl_2_-treated group was lower at any storage time. The DP fruit (1 w + 3 d and 2 w + 6 d 50 μmol/L of the CaCl_2_-treated group) was lower than the control group. Separately, the values were 0.395 ± 0.0225 Ug^−1^FW and 0.391 ± 0.0218 Ug^−1^ FW (Figure 4B). To summarize, PPO activity was higher in DP than in HP. POD activity was HP and far outweighed DP, whether it was the control group or the 50 μmol/L CaCl_2_-treated group (Figure 4C). The HP of the control group had similar changing trajectories as the 50 μmol/L CaCl_2_-treated group HP after 1 w. However, initially, the control group HP was higher than the HP of the 50 μmol/L CaCl_2_-treated group but after the control group HP was lower than the 50 μmol/L CaCl_2_-treated group HP, the values (separately) were 7.85 ± 0.280 Ug^−1^ FW and 17.53 ± 1.075 Ug^−1^ FW. There was no difference with DP; however, the 50 μmol/L CaCl_2_-treated group was slightly higher after 1 w. Both the control group and the 50 μmol/L CaCl_2_-treated group experienced obviously rises. Separately, there were increases from 4.59 ± 0.223 Ug^−1^ FW to 15.33 ± 0.328 Ug^−1^ FW and from 12.64 ± 0.858 Ug^−1^ FW to 25.72 ± 0.170 Ug^−1^ FW.

#### 2.1.5. Antioxidant Enzyme Activities

SOD, CAT, and APX activity were higher in HP than DP for both the control group and the 50 μmol/L CaCl_2_-treated group (Figure 5).

Regarding SOD activity, the control group HP continued rising after 1 and 2 w; the values went from 8.96 ± 0.122 Ug^−1^ FW to 24.90 ± 0.500 Ug^−1^ FW and from 15.33 ± 1.248 Ug^−1^ FW to 31.53 ± 0.373 Ug^−1^ FW (Figure 5A). However, for the 50 μmol/L CaCl_2_-treated group, HP first up and then down for 1 w + 3 d and 2 w + 3 d, reaching the highest point separately at 30.33 ± 2.698 Ug^−1^ FW and 30.87 ± 0.310 Ug^−1^ FW. HP was higher than DP in both the control group and the 50 μmol/L CaCl_2_-treated group at any storage time.

Regarding CAT activity, HP for the control fruit (1 w + 0 d) was higher than the 50 μmol/L CaCl_2_-treated group value, which was 4.08 ± 0.082 Ug^−1^ FW (Figure 5B). For the rest of the storage time, the GP of the 50 μmol/L CaCl_2_-treated group was far more than the control group HP except at 1 w + 3 d. There is no obvious change in DP, but 2 w + 6 d for the 50 μmol/L CaCl_2_-treated group was higher than the control group; the value was 0.810 ± 0.0120 Ug^−1^ FW. HP was higher than DP both in the control group and the 50 μmol/L CaCl_2_-treated group at any storage time.

The APX activity HP 50 μmol/L of the CaCl_2_-treated group was significantly higher than the control group, except at 2 w + 3 d and 2 w + 6 d (Figure 5C). The DP of the 50 μmol/L CaCl_2_-treated group was far more than the control group. The control group HP continuously increased to the highest point: 50.67 ± 0.333 Ug^−1^ FW. The HP of the 50 μmol/L CaCl_2_-treated group rose roughly, except at 2 w + 3 d to the highest point 46.36 ± 0.337 Ug^−1^ FW. HP was higher than DP in both the control group and the 50 μmol/L CaCl_2_-treated group at any storage time.

Regarding the GR activity, the control group HP increased; however, the HP of the 50 μmol/L CaCl_2_-treated group initially decreased and then dramatically increased to 145.18 ± 0.000 UKg^−1^ FW after 1 w (Figure 5D). The control group HP reduced rapidly; the value went from 125.36 ± 3.286 UKg^−1^ FW to 1.18 ± 0.005 UKg^−1^ FW. However, the HP of the 50 μmol/L CaCl_2_-treated group initially increased to the highest point at 92.90 ± 1.390 UKg^−1^ FW and finally decreased after 2 w. In addition, 2 w + 3 d was obviously higher than the control fruit. Moreover, the DP of the 50 μmol/L CaCl_2_-treated group was far more than the control group DP, where at 1 w + 6 d it was 164.90 ± 2.028 UKg^−1^ FW. Regarding DP, there were no obvious changes after 2 w. HP was higher than DP in both the control group and the 50 μmol/L CaCl_2_-treated group at any storage time.

### 2.2. Transcriptome Results

#### 2.2.1. Raw Data Quality Assessment

Table 1 shows the data processing results obtained from the sequencing of pineapple fruit under refrigeration (1 w) and the transfer to room temperature (3 d). The proportion of bases with a quality of no less than 20 after filtration (Q20) was 96–98%, the proportion of bases with a quality of no less than 30 after filtration (Q30) was 91–93%, and the overall data sequencing error rate was 0.03%, and the base numbers of G and C after filtration accounted for 47–52% of the total base number.

#### 2.2.2. Overall Transcriptome Quality Assessment

The principal component analysis (PCA) is shown in Figure 6A; the PCA1 is 89% variance and PCA2 is 5% variance. The CK0d group and TR0d group are indistinct distinctions. In this connection, TR0 d_vs_CK0 d did not conduct the control analysis. The TR3d group and CK3d group were dispersed and each group sample was gathered together.

As shown in Figure 6B, the Pearson correlation coefficient is more than 0.8 after three repetitions. At the same time, the correlation coefficient of the repeated sample is higher than that of the non-repeated sample, which shows that the sample has good biological repeatability, so it can be concluded that the transcriptome sequencing results are reliable. Figure 6C shows each sample’s FAPKM distribution condition.

#### 2.2.3. Screening Results of DEGs

The co-expression Venn diagram (Figure 7A) shows that the differentially expressed gene (DEG) analyses resulted in CK3 d_vs_CK0 d, TR3 d_vs_TR0 d, and TR3 d_vs_CK3 d—three comparison combinations; the total DEGs are 6094, 4167, and 1809, respectively. There were 451 (5.9%) among the number of genes co-expressed.

The volcano map (Figure 7B) shows each comparison combination and the most significant five up- and downregulated genes. Respectively, the CK3 d_vs_CK0 d 5 up- and downregulated genes are the following—upregulated: LOC109718127, LOC109715253, LOC 109718904, LOC 109727840, LOC 109721768; downregulated: LOC 109718469, LOC 109721742, LOC 109704537, LOC 109714910, LOC 109703523. The TR3 d_vs_CK3 d five up- and downregulated genes are the following—upregulated: LOC 109705936, LOC 109722221, LOC 109704995, LOC 109728608, LOC 109719223; downregulated: LOC 109711400, LOC 109725030, LOC 109713516, LOC 109706285, LOC 109718127. The TR3 d_vs_TR0 d five up- and downregulated genes are the following—upregulated: LOC 109721993, LOC 109715253, LOC 109712410, LOC 109708528, LOC 109713189; downregulated: LOC 109703523, LOC 109707759, LOC 109713338, LOC 109714910, LOC 109726213.

The Figure 7C histogram shows the CK3 d_vs_CK0 d, TR3 d_vs_TR0 d, and TR3 d_vs_CK3 d DEG numbers of up- and downregulated genes, respectively, as 3092 and3002,2133 and 2,033,986 and 823.

#### 2.2.4. DEGs, GO, and KEGG Pathway Enrichment Analysis

Gene ontology (GO) is a comprehensive database describing gene functions; through the analysis of DEGs, we found the up- and downregulated GO enrichments most significantly in each comparison combination. From Figure 8A, one can see that the CK3 d_vs_CK0 d-downregulated GO enrichment included plant hormone signal transduction, DNA replication, starch and sucrose metabolism, and cyanoamino acid metabolism. TR3 d_vs_TR0 d-downregulated GO enrichment included plant hormone signal transduction and plant circadian rhythm. Both CK3 d_vs_CK0 d and TR3 d_vs_TR0 d-downregulated GO enrichments have plant hormone signal transductions; however, CK3 d_vs_CK0 d counts more than TR3 d_vs_TR0 d. TR3 d_vs_CK3 d-downregulated GO enrichment may include the biosynthesis of amino acids, glycolysis/gluconeogenesis, and carbon metabolism. For upregulated GO enrichment, CK3 d_vs_CK0 d and TR3 d_vs_TR0 d include the biosynthesis of amino acids, carbon metabolism, alpha-linolenic acid metabolism, tropane, piperidine, and pyridine alkaloid biosynthesis. However, TR3 d_vs_CK3 d was not significantly upregulated in GO enrichment.

As presented in Figure 8B, the Kyoto Encyclopedia of Genes and Genomes (KEGG) shows three comparison combinations CK3 d_vs_CK0 d, TR3 d_vs_TR0 d, and TR3 d_vs_CK3 d (the enrichment results are shown in the figure below). It was found that the above downregulated and upregulated genes were mainly involved in plant hormone signal transduction, starch and sucrose metabolism, biosynthesis of amino acids, carbon metabolism, phenylpropanoid biosynthesis, glycolysis/gluconeogenesis (KEGG ID: obr00010), and the MAPK signaling pathway plant (KEGG ID: obr04016). MAPK signaling pathway is a stress response pathway of plants to the surrounding environment. The MAPK signaling pathway only occurs in CK3 d_vs_CK0 d, proving that the control group can more easily produce ROS compared to the treatment group. In addition, regarding phenylpropanoid biosynthesis occurring in TR3 d_vs_TR0 d, the treatment group can produce more antioxidant bioactive compounds. A further explanation is that the treatment group can alleviate the IB of pineapple.

#### 2.2.5. Screening and Differential Expressions of Oxidation and Antioxidation-Related Genes

Using a heatmap to further prove the relationship between the occurrence of pineapple IB and oxidation and antioxidation systems. A total of 83 DEG-related genes were screened from transcriptome data, including POD (36), SOD (8), APX (5), GSH-PX (3), GR (1), CAT (3), PPO (4), and GST (22), respectively. The APX family (g.109727824, g.109726514), CAT family (g.109729040, g.109707674), SOD family (g.109716880), and GR (g.109727484), respectively, were upregulated after fruit disease. The PPO family (g.109719920, g.109719939) was upregulated. The GRX 4 family (g.109711896) upregulated during the treatment group (3 d). Many genes of the GST family and POD family were upregulated under fruit disease. Perhaps these are related to the occurrence of IB. These phenomena show the pineapple IB is related to oxidation and antioxidation systems. Pineapple fruit will activate its own antioxidant system to eliminate ROS, but the oxidase activity will also increase after browning. These are also consistent with the physiological results (Figure 9).

#### 2.2.6. Phenylpropanoid Biosynthesis and MAPK Pathway

The plant shikimate pathway is the entry to the biosynthesis of phenylpropanoids. The pathway of shikimic acid involves producing aromatic amino acids and other aromatic compounds (phenylpropanoids) (Figure 10C). Seventy-four DEGs were detected in this pathway, including cytochrome (CYP), 4-coumarate-CoA ligase (4CL), putrescine hydroxycinnamoyl transferase(PHT), peroxidase (POD), β-glucosidase, cationic peroxidase (Cat-POD), caffeoyl shikimate esterase (CSE), trans-cinnamate 4-monooxygenase (T-C4M), phenylalanine ammonia-lyase-like (PAL), hydroxy cinnamoyl transferase(HCT), cinnamyl alcohol dehydrogenase (CAD), BAHD acyltransferase (BAHD-acylt), caffeoyl-CoA O-methyltransferase (CCoAOMT), aldehyde dehydrogenase (ALD), putrescine hydroxycinnamoyl transferase (PHT), vinorine synthase (VSY), salutaridinol 7-O-acetyltransferase (SOAT), and cinnamoyl-CoA reductase (CCoAR). PAL, 4CL, and HCT upregulation can lead to CAD upregulation and will increase the lignin content. During the heatmap analysis, PAL (g.109727686, g.109707567) was upregulated in the control and treatment groups at 3 d. Moreover, 4CL (g.109720819, g.109711909) and HCT (g.109703812, g.109706886) upregulation led to CAD (g.109715017) upregulation. Upregulation of these enzyme genes will increase lignin synthesis (Figure 10D). Lignin biosynthesis (Figure 10E) is a cross-linked phenolic polymer. Therefore, total phenol content will enhance. It can be seen from the physiological results in Figure 4 that the total phenol content at 1 w + 3 d is higher than 1 w + 0 d, which is consistent with the transcriptome data.

The MAPK pathway (Figure 10A) is mainly related to pathogen infection, pathogen attack, phytohormones, external environmental stimulation (cold, salt, drought, osmotic stress, etc.), ozone, and wounding. Under a series of enzymatic reactions, H_2_O_2_ can further cause cell death and the H_2_O_2_ product. ABA can increase fruit stress adaptation and CAT can suppress the H_2_O_2_ product increase regarding the fruit stress-tolerant response. Furthermore, Ca^2+^ can regulate ROS homeostasis. As seen in Figure 10B, the heat map analysis of the MAPK pathway chitinase is highly expressed in HP (g.109713885). At the same time, the expression in the TR group is significantly more than in the CK group, indicating that the disease in the control group is more serious. The WRKY gene family is also related to plant resistance; g.109709450, g.10970389, and g.109719592 are highly expressed in DP. The expression of the CK group was significantly more than that of the TR group, indicating that the expressions of WRKY gene family genes would increase when plants were stimulated by the outside world.

Plant cells are affected by many factors, Figure 10F shows that phenylpropanoid biosynthesis and the MAPK pathway regulate ROS when plant cells are under cold stress. Under cold stress stimulation, secondary signal molecules are induced on the cell membrane. Phenylpropanoid biosynthesis mainly produces antioxidant compounds, which turn H_2_O_2_ and O_2_^• −^ into H_2_O; O_2_ will maintain ROS homeostasis. However, H_2_O_2_ and O_2_^• −^ accumulation will cause cell death and resistance response. MAPK regulated Ca^2+^ retention of ROS homeostasis sometimes regulated ABA stimulation; MAPKKK, MAPKK, MAPK product H_2_O_2_, and CAT1 can inhibit the H_2_O_2_ product and make cells resist external stress. ROS homeostasis plays an important role in cells, out-of-balance ROS will cause cell death. Pineapple IB may be closely related to ROS homeostasis.

#### 2.2.7. Heatmap Analysis of Related Pathways

Regarding plant hormone signals (Figure 11A), including ABA, ET, JA, and IAA gene expression, originally, auxin first induces early response gene expressions, including AuxIAA, GH3, and SAUR. AuxIAA (g.1097117751, g.109703565, g.109708445, g.109707217, g.109716520, g.109727231, g.109712321, g.109710481), GH3 (g.109724502), and SAUR (g.109707214) are upregulated in HP. Contrarily, AuxIAA (g.109716736), GH3 (g.109723185, g.109712463, g.109708627), and SAUR (g.109728554, g.109707017) are upregulated in DP. JA generates JA-IIe by catalysis of JAR; JA-IIe will increase the defense abilities of plants. JAR1(g.109716181) is upregulated in DP, and the CK group is more than the TR group; the sideline CaCl_2_ treatment can increase disease resistance. On the contrary, JAR2 (g.109714846, g.109717217) upregulated in HP. In the process of the ABA signal transduction, in HP, both PYR (g.109713936, g.109705333) and PYL (g.1097,10912, g.109717818) are upregulated, leading to PP2C (g.109716365) and Snpk2 (g.109705826, g.109714251) upregulation, promote downstream gene expression, open the ABA signaling pathway, and regulate plant stress tolerance or growth and development. However, PYL (g.109717557), PP2C (g.109708133, g.109709262, g.109727986), and Snpk2 (g.109712467, g.109727308) are upregulated in DP. Moreover, the CK group genes are highly expressed. ETR3 (g.109712816) gene mass expression lead to EIN2 (g.109707756) upregulation, causing a series of downstream responses. 

The plant–pathogen interaction (Figure 11B) includes the antiviral protein, pathogenic protein, heat shock protein, respiratory burst oxidase, and cysteine protease. Respiratory burst oxidase can produce a little of ROS in the plant body to resist external pressure. Genes (g.109722976, g.109713174) are great quantity expressions in DP, g.109717636 is expressed in HP. Cytochrome oxidase (g.109703608) promotes ATP synthesis in DP (Figure 11E).

The tricarboxylic acid cycle (TAC) is the second stage of aerobic respiration (Figure 11F) that shows pyruvate dehydrogenase (g.109721645, g.109728788, g.109719561), promotes acetone to produce acetyl coenzyme A, generates citric acid, and enters the citric acid cycle after fruit morbidity. During gluconeogenesis, oxaloacetic acid generates phosphoenolpyruvate and carbon dioxide under the action of phosphoenolpyruvate carboxylation kinase (g.109710186, g.109714503) in DP. TR group genes express more than the CK group.

Figure 11C,D,G show that fatty acid biosynthesis, fatty acid biodegradation, and fatty acid metabolism play important roles in the cold resistance of plants. In the process of fatty acid synthesis, acetyl coenzyme A generates malonyl coenzyme A (g.109713239) under the action of acetyl coenzyme A carboxylase (ACC) (g.109722793) in DP (Figure 11C). Acyl carrier protein (ACP) is a domain in the fatty acid synthase (FAS) peptide chain, which is used to synthesize fatty acids. ACP (g.109723533, g.109726849, g.109714538) is upregulated in DP, but ACP (g.109713115) is in HP. Peroxisomal (g.109716412) β-oxidation mainly oxidizes ultra-long-chain fatty acids (Figure 11D), Fatty acyl coenzyme A with a shortened chain is formed and then transported to mitochondria for complete oxidation. 

Fatty acid oxidation also takes place in other organelles, such as endoplasmic reticulum ω-oxidation and the introduction of hydroxyl groups on the last carbons of fatty acids, followed by oxidation to aldehyde and carboxyl groups to form dicarboxylic acids. The enzymes involved are ω-hydroxylase (g.109721150), alcohol dehydrogenase (g.109707359, g.109721267), and aldehyde dehydrogenase (g.109728684, g.109723285), which are also involved in the metabolic process of ethanol (Figure 11G).

#### 2.2.8. Transcription Factor Analysis

Among all of the gene families, we selected the 15 gene families with the largest proportions (Figure 12). The six families with the largest proportions are Pkinase (244), Pkinase Tyr (134), P450 (79), Myb DNA-binding (60), NB AR (59), and PPR (46). Both Pkinase and Pkinase Tyr are related to the MAPK pathway. When plants are under adverse conditions, MAPKK and MAPK can be activated in turn and jointly regulate cellular physiological and pathological processes, mainly receptor proteins, protein kinase, and signal transductions. P450 main includes cytochrome, premnaspirodiene oxygenase, oryzalexin E synthase, abscisic acid 8′-hydroxylase, geraniol 8-hydroxylase, Ent-kaurenoic acid oxidase, alkane hydroxylase MAH, isoflavone e 2′-hydroxylase, allene oxide synthase, flavonoid 3′-monooxygenase, 3,9-dihydroxypterocarpan 6A-monooxygenase, and trans-cinnamate 4-monooxygenase. NB_AR is mainly related to disease resistance proteins. The heatmap of these gene families is shown in Figure 13.

Calmodulin-binding receptor-like cytoplasmic kinase 2 (CRCK2) can against various diseases. From Figure 13A,B, Calmodulin-binding receptor-like cytoplasmic kinase 2 (g.109712591, g.109728905, g.109727024, and g.109725844) is upregulated in DP (3 d), but g.109727204 and g.109718861 are upregulated in HP (0 d). Calcium-dependent protein kinase (g.109727237, g.109714958, g.109715490, g.109715759) is upregulated in DP (3 d). Compared with the CK group, the FPKM of the TR group are lower. It shows that CaCl_2_ is effective in fruit disease. Flavonoid 3′-monooxygenase-like (F3’H) belongs to the cytochrome P450 subfamily, which catalyzes NADPH and the O_2_-dependent monooxygenation. Anthocyanins and proanthocyanidins can be produced, which can prevent cells from being damaged by strong UV-B radiation. Flavonoid 3′-hydroxylase can also participate in the acetic malonic acid pathway through flavanone substances to protect plants from animals, plants, pathogenic microorganisms, etc. F3’H (g.109720891, g.109721770) is upregulated in HP (0 d) (Figure 13C). It shows that the content of anthocyanins and other antioxidant substances decreases after fruit disease. Allene oxide synthase (AOS) generates precursors of the defense hormone jasmonate (JA). AOS (g.109720968) is upregulated in DP (3 d). It is proven that a large number of resistance hormones will be synthesized after fruit disease. The MYB TF family refers to a class of transcription factors containing the MYB domain, which is a peptide of about 51–52 amino acids and contains a great deal of highly conserved amino acid residues and spacer sequences. The first involves tryptophan residues with regular intervals of about 18 amino acids, which participate in the formation of hydrophobic cores in the spatial structure (Figure 13D). From (Figure 13E), the largest proportion of the resistance protein is upregulated in HP (0 d). However, the TR group DP FPKM is further than the CK group. The pentatricopeptide repeat-containing protein is the largest protein family in plants for plant growth development (Figure 13F).

## 3. Discussion

The pineapple, which is a tropical or sub-tropical fruit, is usually transported by a cold chain process [23]. Refrigeration is a general method used to store fruits and vegetables, but cold stress can induce the chilling injury of pineapple fruit, which principally restricts the postharvest fruit quality and shelf life of pineapples [24]. To date, plenty of physical and chemical tests have shown good progress in delaying pineapple CI. However, it is difficult to bring about massive applications for economic costs, the stability of efficacy, and operation accessibility; therefore, developing more practical and reliable approaches to ameliorate the storability of pineapple fruit is (as usual) required.

Ca, as a second messenger, is important in plant cold stress response. Studies have shown that CaCl_2_ treatment alleviated bermudagrass chilling injury by regulating reactive oxygen species (ROS) of products and lessening cell damage [25]. In addition, whether it involves pre- and postharvest foliar application calcium, the effects of delayed aging or ripening and improving physiological disorders are apparent in fruits and vegetables [26]. Ca can improve apple fruit firmness and reduce fruit decay [27]. CaCl_2_ can reduce the occurrence of the internal browning of pineapple fruit under low temperatures [28]. Our present study also obtained similar results. 

In the present study, pineapples were processed via exogenous CaCl_2_ (50, 100, and 150 μmol/L) under chilling temperatures at different times; they were transferred to a normal temperature to simulate the sale of pineapple fruit in actual production. We investigated what affects the chilling temperature, inducing the CI development of post-harvest pineapple via exogenous CaCl_2_ treatment and no treatment. To screen for the best CaCl_2_ concentrations, we measured some characterization indicators and related indexes of cell membrane destruction, such as the incidence rate, IB index, and electrical conductivity. A large amount of research has confirmed that CaCl_2_ can alleviate CI (the incidence rate, IB index, and electrical conductivity) [29]. In our present study, after the transfer from refrigeration to room temperature, the incidence rate, IB index, and electrical conductivity increased, indicating that their membranes had become damaged. Previous studies have proved that CaCl_2_ treatment can alleviate CI symptoms. Meanwhile, along with the extension of storage time, the lower the temperature, the more serious the chilling injury [30]. In our study, it was not hard to find similar results. It was proved (again) that calcium can enhance cold tolerance in pineapple fruit.

Low-temperature induction of reactive oxygen species (ROS), such as O_2_^• −^ and H_2_O_2_, in turn, caused cell damage and resulted in chilling symptoms [31]. In our study, certain similarities changed between the ROS (O_2_^• −^ and H_2_O_2_) and MDA of fruit DP, showing the accumulation of lipid peroxide and enhanced membrane permeability after fruit disease [32]. 

The ascorbate-glutathione (AsA-GSH) cycle plays an important role in the antioxidant pathway, which can directly scavenge the H_2_O_2_ produced. Moreover, ASA and GSH are the most abundant low molecular non-enzymatic antioxidants and GR is a key enzyme in the AsA-GSH cycle [33]. Some studies have shown that CaCl_2_ treatment alleviated CI by enhancing the AsA-GSH cycle and antioxidant enzyme activities to quench ROS in loquat fruit [34]. In the present study, AsA-GSH content and APX and GR activity distinctly decreased in DP, indicating that antioxidant capacity is not enough to resist external cold stress, leading to the IB of pineapple fruit.

During storage, pineapple fruit CI browning symptoms were a result of a physiological disorder, which is usually due to oxidation of phenolic compounds and accumulation of polyphenol oxidase [35]. In the present study, there was PPO activity contrary to the POD activity of HP and DP; however, the POD activity changing tendency was similar to other antioxidant enzymes, which were enhanced during cold storage and decreased strongly at room temperature [36]. This indicates that POD is a two-way street enzyme, which may serve as both oxidase and antioxidant enzymes.

ROS was reflected in the total antioxidant capacity and APX activity [32]. ROS homeostasis is usually maintained by the balance of ROS production and scavenging in plant organisms. Plant antioxidant systems—to eliminate ROS—including enzymatic and non-enzymatic antioxidative components (to maintain ROS homeostasis) [37]. Antioxidative enzymes, such as SOD, APX, and CAT, play important roles in the ROS -scavenging pathway [37]. In our study, there was a rapid increase in the activities of the enzymatic antioxidative biomarkers (SOD, APX, CAT) by CaCl_2_ treatment. Furthermore, the activities of enzymatic antioxidative biomarkers decreased in the DP of fruit. It was confirmed (on various fruits) that CaCl_2_ treatment maintained higher activities of enzymatic antioxidative biomarkers [38]. 

Through the transcriptome data analysis, we concluded that fruit disease was related to oxidation and oxidation resistance. Phenylpropanoid biosynthesis is related to the synthesis of phenolic substances. Total phenols, flavonoids, and anthocyanins are antioxidant bioactive compounds [39]. In this study, the upregulation of genes related to lignin synthesis led to the increase in total phenol content, which is consistent with the physiological results. The MAPK pathway is a stress response of plants to the outside world and is related to ROS. It has been proven that ROS can induce MAPK activation; meanwhile, MAPK cascades can modulate ROS production and responses [40]. By the transcription factor analysis, we can assume that the two largest families are related to MAPK and the plant disease resistance protein. It may be mainly related to the occurrence of the IB of pineapple.

## 4. Materials and Methods

### 4.1. Plant Material and Treatments 

Mature green stage pineapple (*Ananas comosus* L. Merr.) was harvested from a local orchard in Qionghai city, Hainan Province, and transferred to a laboratory at the Hainan University Food Science and Engineering building within 3 h. A total of 600 fruits were randomly selected, with 150 fruits in each group (uniform size and color). All fruits were cleaned, disinfected, and then dried for the next step. Three groups were used, separately immersed in 50, 100, and 150 μmol/L of CaCl_2_ solution in 100 L of deionized water for 5 min at 25 °C. One group was dipped in deionized water for 5 min at 25 °C and was served as the control. After drying with natural air, fruits were packed in a plastic box with plastic wrap for three hours to ensure full absorption. Then, all of the fruits were transferred to cold storage (6 ± 1 °C and 85 ± 5% relative humidity (RH)) for subsequent experiments.

After cold storage for 1 w, we divided all processed fruits into two parts—one part was transferred to normal temperature (25 °C ± 1 °C and 85 ± 5% relative humidity (RH)) and the remainder of the fruits did not move from the cold storage for 2 w. The pulps near the cores of the pineapples were taken for experimental study. Fifteen fruits were sampled from each treatment after cold storage (1 w) and were transferred to room temperature at 0, 3, and 6 d. There were three replicates included in each treatment. Similarly, the fruit samples obtained after cold storage (2 w) were transferred to room temperature at 0, 3, and 6 d. According to the statistical results of internal browning, three pineapple fruits with similar statistical conditions were selected from each group, and the flesh “F/C (Flesh/core)” parts were taken, respectively. One part was used to determine the physiological indexes, and the other part was immediately mixed with liquid nitrogen and stored in a −80 °C ultra-low-temperature refrigerator for reserve. Meanwhile, all of the samples (ground to powder under liquid nitrogen) were stored at −80 °C for subsequent experiments.

### 4.2. Optimum Concentration Obtained

#### 4.2.1. IB Appearance

The fruit was cut in half and the IB appearance was photograph-recorded during each sampling.

#### 4.2.2. Electrical Conductivity Measurement

Relative conductivity involved the eigenvalue index reflecting membrane permeability. Methods refer to [41] and were slightly modified. Nine fruits were cut in half—half were used for conductivity determination and half for sampling. A hole punch (6 mm in diameter) was used randomly, with 5 little discs picked from the pulp near the heart of the fruit, for a total of 15 little discs. Then they were cleaned 3 times with distilled water and rocked for half an hour in the water bath. The weight and temperature remained unchanged before and after boiling for 20 min. Meanwhile, the initial (E_0_) and final (E_t_) values were measured using a conductivity meter before and after boiling. The formula is as follows: $$\mathrm{Relative}\;\mathrm{conductivity}\;\left(\%\right)=\frac{E0}{\mathrm{Et}}\times 100\;\%$$

### 4.3. Determination of Physiological and Biochemical Indexes

#### 4.3.1. Assays of ROS Production and Membrane Lipid Peroxidation MDA Contents

Malondialdehyde (MDA) is one of the membrane lipid peroxidation products determined by the thiobarbituric acid assay [41], slightly modified. Accumulation caused damage to the cytoplasmic membrane and organelles of fruit thinning cells. Moreover, 0.1 g of flesh was added to 1 mL of 100 g/L of TCA and centrifuged at 10,000× *g* at 4 °C for 20 min. We took the supernatant and added 2.0 mL of 0.67%TBA boiled in boiling water for 20 min, cooling the centrifugation again. Then the absorbance value was measured at 450,532,600 nm. The MDA content was calculated by formula.

The ROS production was assessed by the Superoxide Anion Radical (O_2_^• −^) generation rate and the hydrogen peroxide (H_2_O_2_) content was determined by using kits from Suzhou Grace (China). 

#### 4.3.2. Determination of Total Phenolic Content

Polyphenols play important roles for plants because of their antioxidant properties. We followed the method in [42], slightly modified. We weighed 1 g of the powdered sample and added it to 80% pre-cooled methanol solution at 4 °C for 20 min after oscillation blending. We took 500 µL of extract, added 4 mL of pre-cooled methanol solution, and 500 µL of Folin phenol, and let it stand for 5 min. Then we added 5 mL of 20% sodium carbonate solution in darkness at 25 °C for 1 h. The total phenol content was calculated according to the gallic acid standard curve.

#### 4.3.3. Assessments of Polyphenol Oxidase (PPO) and Peroxidase (POD) Activities

For polyphenol oxidase (PPO) [43] and peroxidase (POD) [44], slightly modified, frozen tissues (0.1 and 0.3 g) were added to 2 mL of acetate buffer solution (1 mM PEG, 4% PVP, and 1% Triton X-100), homogenized at 4 °C, and centrifuged at 12,000× *g* at 4 °C for 30 min to extract the enzyme solution. 

PPO and POD were determined by a UV-5000PC spectrophotometer, METASH, China. Respectively, absorbances were at 420 and 470 nm. The cuvette was added to 1.6 mL of 50 mM of phosphate buffer (pH 5.5), 0.4 mL of 10 mM pyrocatechol, and then quickly added 0.2 mL of enzyme extract solution, blending the measure absorbance change curve. One unit (U) of PPO activity was defined as the amount of the enzyme that caused a change in A_420_ of 0.01 min^−1^. The cuvette was added to 2 mL of 25 mM of guaiacol, 0.2 mL of H_2_O_2_, and 0.1 mL of an enzyme extract to measure POD activity. One U of POD activity was defined as the amount of the enzyme that caused an increase in the A_470_ of min^−1^.

#### 4.3.4. Assessments of Ascorbic Acid (ASA) and Glutathione (GSH) Contents

ASA and GSH play pivotal roles in the AsA-GSH cycle. Glutathione regenerates ASA in the AsA-GSH cycle and regulates H_2_O_2_ [33]. The ASA content was determined by the red phenanthroline method [45], slightly modified. We took 0.1 g of pulp and added 1 mL of TCA solution, extracted for 10 min, and centrifuged at 12,000× *g* at 4 °C for 30 min. Then we took 0.5 mL of the extract, added 50 g/LTCA and 1 mL of anhydrous ethanol in turn, and mixed. Finally, 0.5 mL of 0.4% phosphoric acid-ethanol solution, 1 mL of 5 g/LBP- ethanol solution, and 0.5 mL of 0.3 g/L FeCl_3_-ethanol solution were successively added. The absorbance value was measured at 534 nm after 60 min of a water bath at 30 °C. The content of the ascorbic acid was calculated by the standard curve.

The content of GSH was determined by the dithio-dinitrobenzoic (DTNB) acid method [46], slightly modified. We added 0.1 g of flesh to 1 mL of precooled 50 g/L trichloroacetic acid solution (containing 5 mmol/EDTA-Na2) and centrifuged at 12,000× *g* at 4 °C for 20 min. We took two test tubes and added 0.5 mL of supernatant and 1.0 mL of 0.1 mol/L pH 7.7 phosphoric acid buffer solution, respectively. Then, to one tube, we added 0.5 mL of 4 mmol/LDTNB solution; to another tube, we added 0.5 mL of 0.1 mol/L pH 6.8 phosphoric acid buffer solution. Two test tubes were placed at 25 °C (thermal reaction) for 10 min. The content of the GSH was calculated by the standard curve.

#### 4.3.5. Assessment of Antioxidant Enzyme Activities

Enzyme activity was conducted by the kinetic analysis of the absorbance value. The determination of GR [47]: 0.1 g of flesh was added to a 5.0 mL, 4 °C precooled, 0.1 mol/L pH 7.5 phosphoric acid buffer solution (containing 1 mmol/EDTA-Na2) and centrifuged at 12,000× *g* at 4 °C for 20 min. Then, we added a 0.9 mL 0.1 mol/L pH 7.5 phosphoric acid buffer solution (containing 1 mmol/EDTA), 0.34 mL 5 mmol/L GSSG solution, and 0.067 mL enzyme extract solution. Finally, 13.33 μL of 4 mmol/L NADPH solution was added to initiate the enzyme reaction. We immediately measured the dynamic OD value every 5 s for 35 s at 340 nm. The enzyme activity was calculated according to the formula. The determination of APX refers to [48], slightly modified. Flesh powder (0.1 g) was used to analyze each APX activity. The APX activity was presented by U kg^−1^ FW.

Catalase activity (CAT) and superoxide dismutase (SOD) were determined by using kits from Suzhou Grace (China). 

### 4.4. Transcriptome Data Analysis

#### 4.4.1. Transcriptome Data Quality Assessment

The transcriptome data quality were evaluated by analyzing the transcriptome raw data, principal component analysis (PCA), pearson correlation coefficient and boxplot, the transcriptome data quality Respectively, raw data was assessed by Er%, Q20%, Q30%, and GC% in the proportion of the total base number. The principal component analysis (PCA), Pearson correlation coefficient, and FPKM value boxplot that were analyzed show the correlation between samples.

#### 4.4.2. GO and KEGG Enrichment Analysis

GO is a database of gene function information; it is divided into three types: biological process (BP), cellular component (CC), and molecular function (MF), and uses less than 0.05 padj as the threshold of significant enrichment.

KEGG is a database related to metabolic pathways and uses less than 0.05 padj as the threshold of significant enrichment.

#### 4.4.3. Screening of Related DEGs, Analysis of Related Pathways, and Transcription Factor Analysis

We screened the genes related to physiological indicators through GO and KEGG annotation analysis and then analyzed the expressions of the screened genes in each sample by a heatmap. We found the metabolic pathway related to this experiment by consulting, analyzing the genes in the pathway by the heat map, excavating the material change information in the metabolic pathway, and obtaining the conclusions related to this experiment. We analyzed the transcription factors of all genes, counted the top 15 transcription factors with the largest proportions, and made a pie chart. In addition, we made a heat map analysis of the genes of the largest six transcription factor families.

## 5. Conclusions

In the present study, pineapple IB was related to ROS homeostasis, which included the organism’s relative balance between enzymatic and non-enzymatic antioxidants and oxides. Physiological data and transcriptome data also support this triangulation. The antioxidant capacities (SOD, POD, CAT, APX, and GR) decreased and the oxidation capacity (PPO) increased after fruit browning. Moreover, pineapple fruit treated with CaCl_2_ improved this ‘out-of-balance’. However, this change is difficult to understand through physiological indicators. Therefore, we analyzed it at the molecular level through the transcriptome. With pineapple IB, some metabolic pathway genes will be regulated, including phenylpropanoid biosynthesis and the MAPK pathway. These transcriptome analyses provide us with many relevant gene networks, which are very helpful for us to solve pineapple IB at the molecular level by genetic engineering.

## Figures and Tables

**Figure 1 plants-11-02215-f001:**
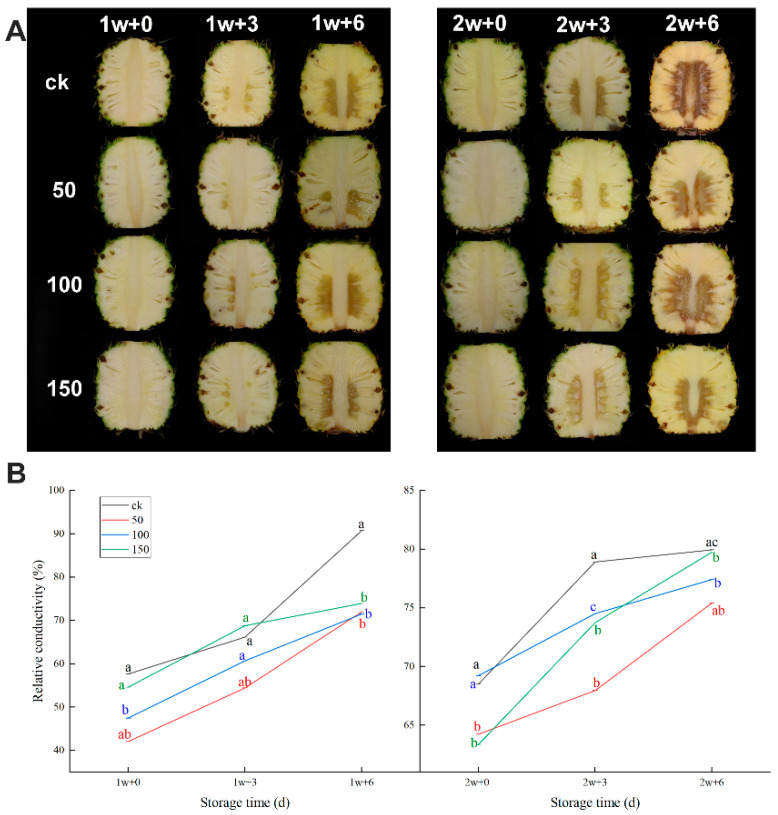
IB appearance (**A**), electrical conductivity (**B**) in pineapple fruit during storage at 6 °C (1 and 2 w) after transfer to room temperature (3 and 6 d), and treatment with CaCl_2_ (50, 100, 150 μmol/L) or water (CK). Vertical bars represent the standard error of the mean of triplicate assays. Characters at the same storage times represent significant differences (*p* < 0.05).

**Figure 2 plants-11-02215-f002:**
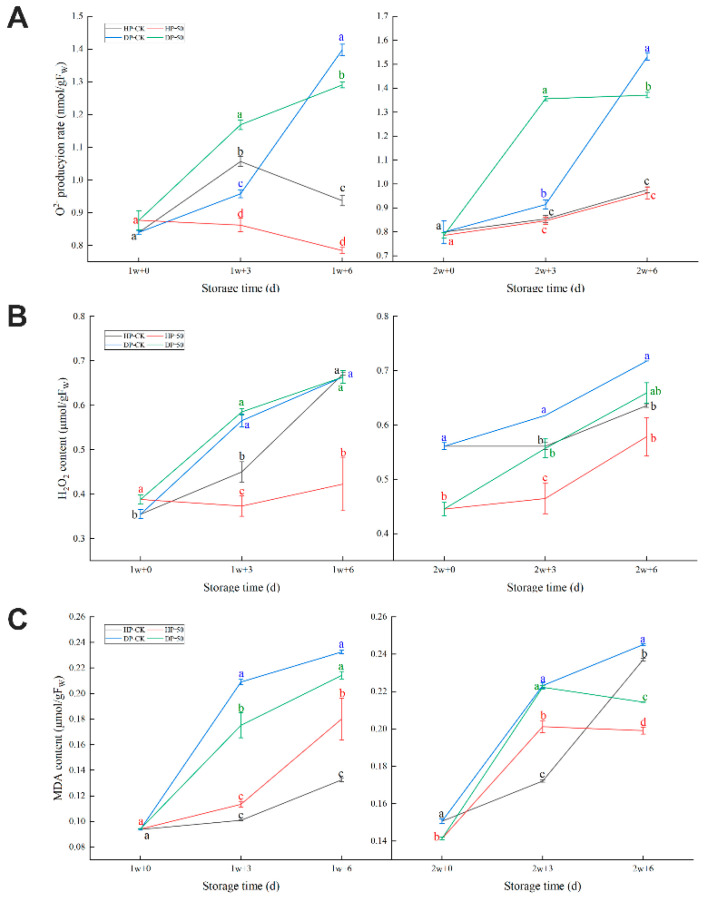
O_2_^• −^ production rate (**A**), H_2_O_2_ content (**B**), and MDA content (**C**) in pineapple fruit during storage at 6 °C (1 and 2 w) after transfer to room temperature (3 and 6 d), and treatment with CaCl_2_ (50 μmol/L) or water (CK). HP: “health part”, DP: “disease part”. Vertical bars represent the standard error of the mean of triplicate assays. Characters at the same storage times represent significant differences (*p* < 0.05).

**Figure 3 plants-11-02215-f003:**
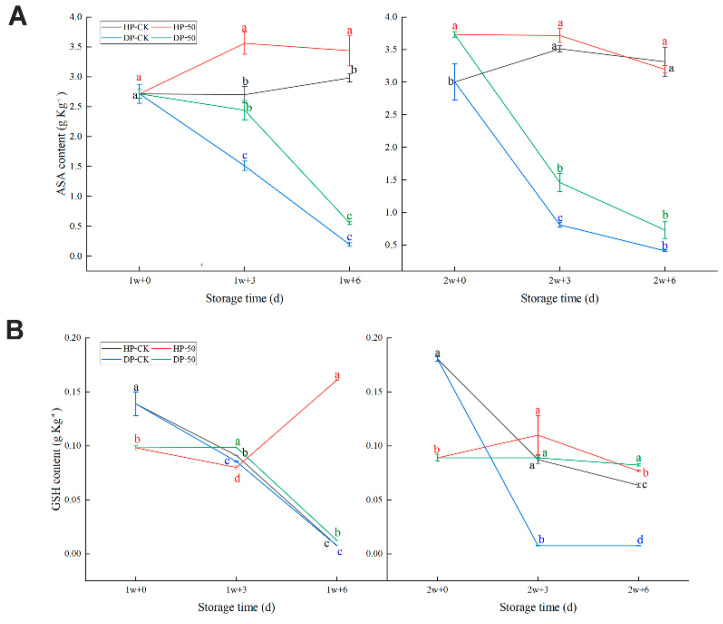
ASA content (**A**) and GSH content (**B**) in pineapple fruit during storage at 6 °C (1 and 2 w) after transfer to room temperature (3 and 6 d), and treatment with CaCl_2_ (50 μmol/L) or water (CK). HP: “health part”, DP: “disease part”. Vertical bars represent the standard error of the mean of triplicate assays. Characters at the same storage times represent significant differences (*p* < 0.05).

**Figure 4 plants-11-02215-f004:**
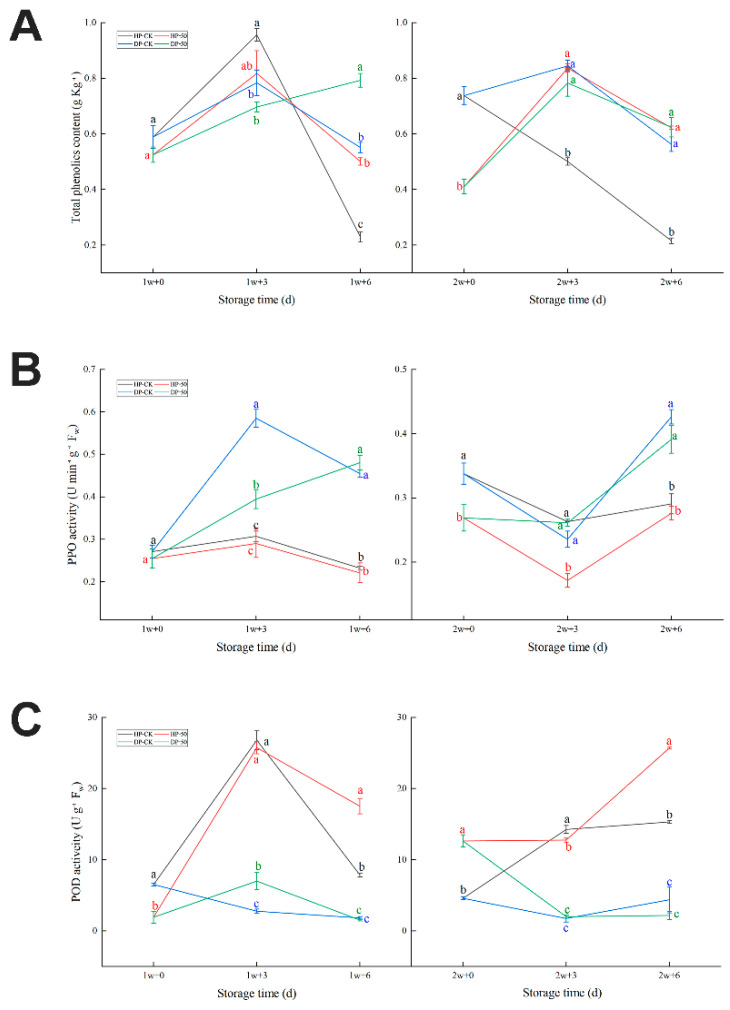
Total phenolics content (**A**), PPO activity (**B**), and POD activity (**C**) in pineapple fruit during storage at 6 °C (1 and 2 w) after transfer to room temperature (3 and 6 d), and treatment with CaCl_2_ (50 μmol/L) or water (CK). HP: “health part”, DP: “disease part”. Vertical bars represent the standard error of the mean of triplicate assays. Characters at the same storage times represent significant differences (*p* < 0.05).

**Figure 5 plants-11-02215-f005:**
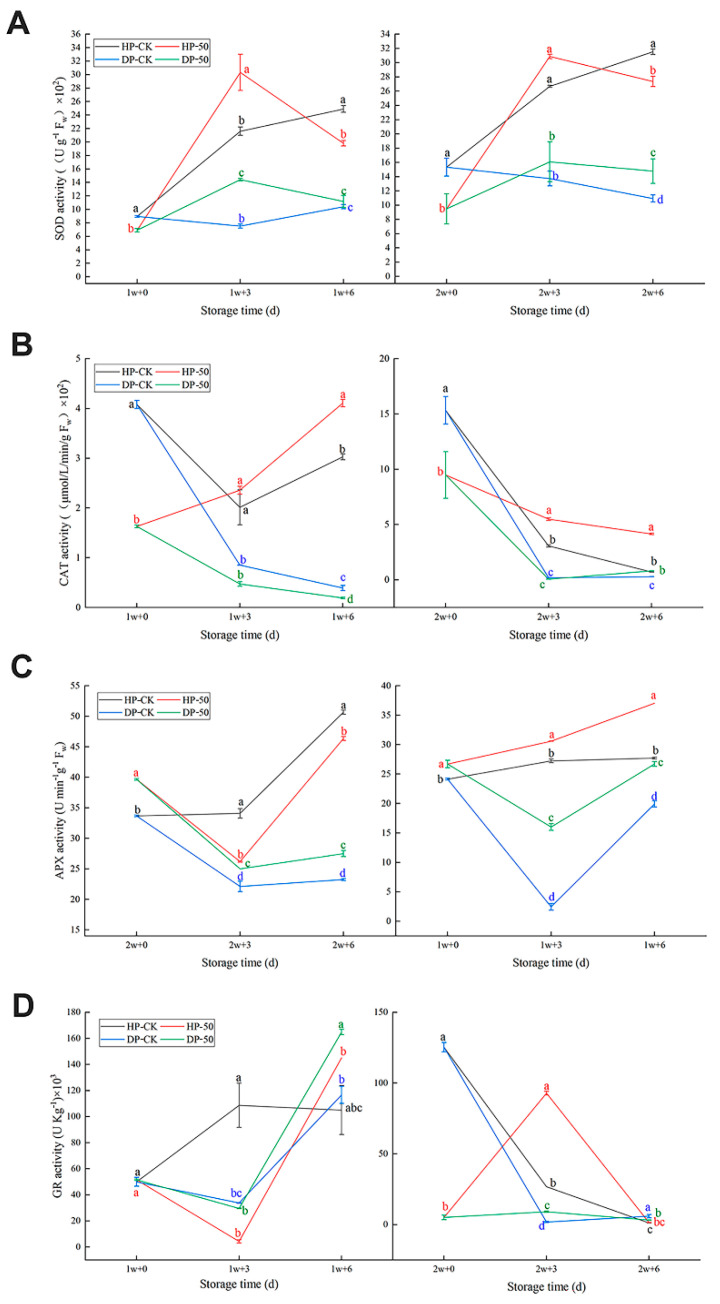
SOD activity (**A**), CAT activity (**B**), APX activity (**C**), and GR activity (**D**) in pineapple fruit during storage at 6 °C (1 and 2 w) after transfer to room temperature (3 and 6 d), and treatment with CaCl_2_ (50 μmol/L) or water (CK). HP: “health part”, DP: “disease part”. Vertical bars represent the standard error of the mean of triplicate assays. Characters at the same storage times represent significant differences (*p* < 0.05).

**Figure 6 plants-11-02215-f006:**
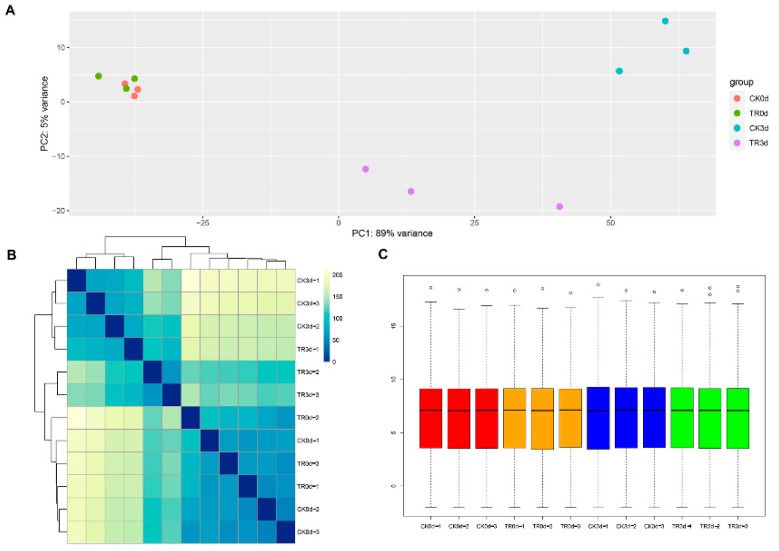
Principal component analysis (PCA) (**A**), Pearson correlation coefficient (**B**), and Boxplot for FPKM values (**C**). Boxplot C: Red represents”CK0d”; Orange represents”TR0d”; Blue represents”CK3d”; Green represents”TR0d”.

**Figure 7 plants-11-02215-f007:**
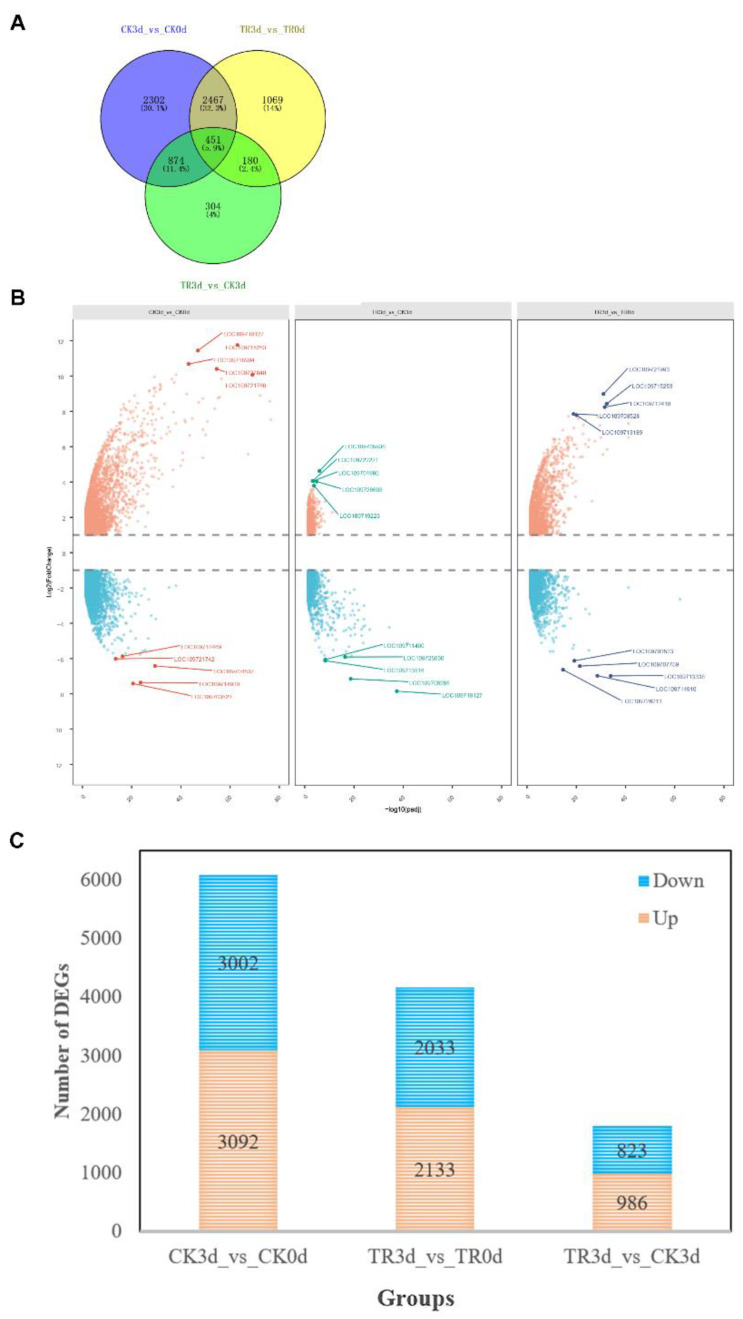
Venn diagram, volcano map, and histogram of the differentially expressed genes (DEGs), comparing the pineapple fruit health and disease parts on different days. (**A**) The Venn diagrams showing similarly or distinctly regulated genes within the control and treatment groups at 0 d and 3 d. (**B**) The volcano map showing the Log2 (fold change) upregulated or downregulated in each comparison combination; the five most significant upregulated and downregulated genes were selected. (**C**) The histogram showing the number of DEGs upregulated or downregulated in each comparison combination.

**Figure 8 plants-11-02215-f008:**
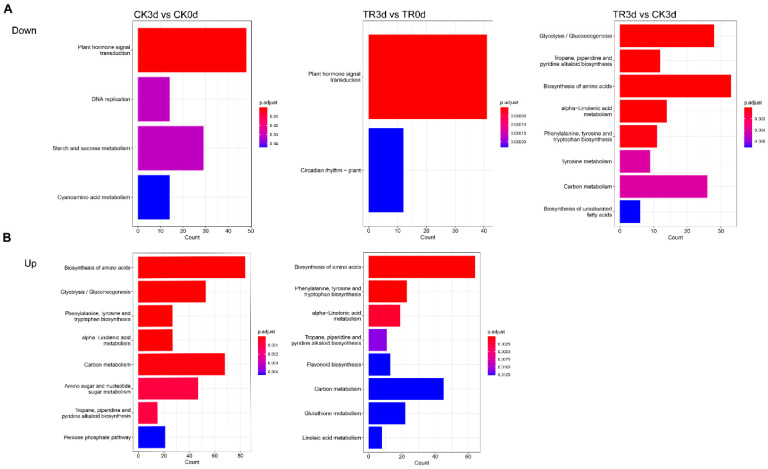
GO and KEGG—the upregulated and downregulated genes in each comparison combination are classified, respectively. (**A**,**B**) GO (gene ontology) Down and Up histogram, a comprehensive database describing gene functions. (**C**,**D**) KEGG (Kyoto Encyclopedia of Genes and Genomes) Down and Up point diagram, a comprehensive database integrating genomes, chemistry, and system function information.

**Figure 9 plants-11-02215-f009:**
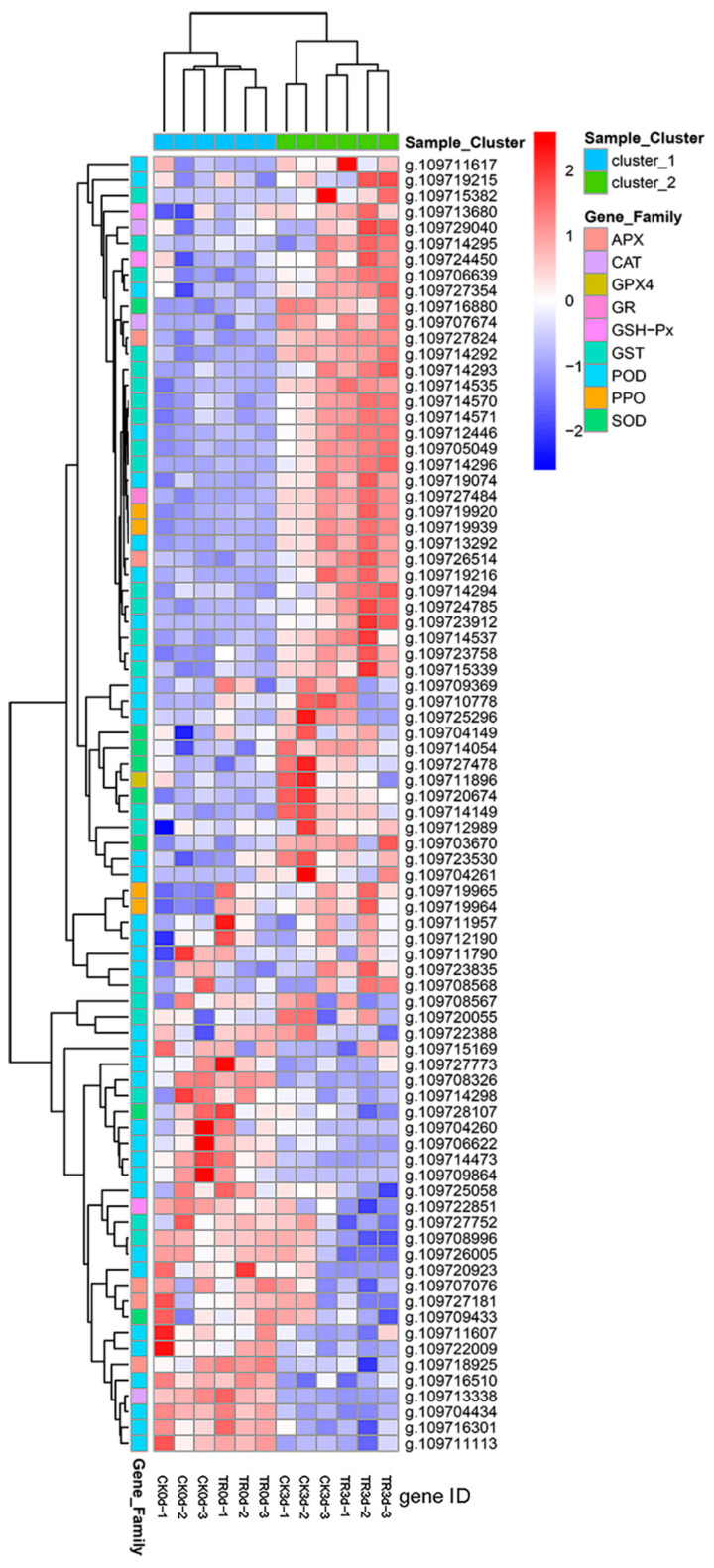
Differential expressions of oxidation and antioxidation-related genes heatmap.

**Figure 10 plants-11-02215-f010:**
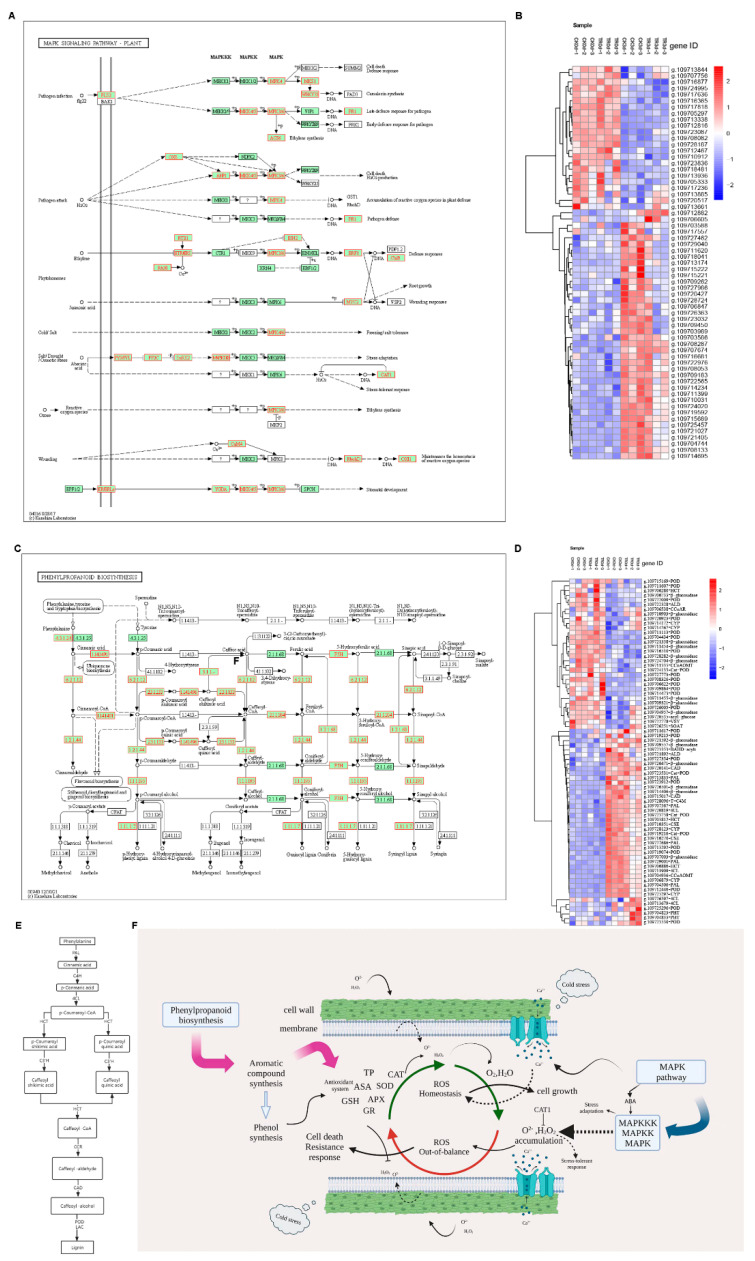
(**A**) MAPK pathway. (**B**) Heatmap of the MAPK pathway. (**C**) Phenylpropanoid biosynthesis. (**D**) Heatmap of phenylpropanoid biosynthesis in each comparison combination. (**E**) Lignin biosynthesis; metabolic tree (**F**) Phenylpropanoid biosynthesis and the MAPK pathway are related to ROS.

**Figure 11 plants-11-02215-f011:**
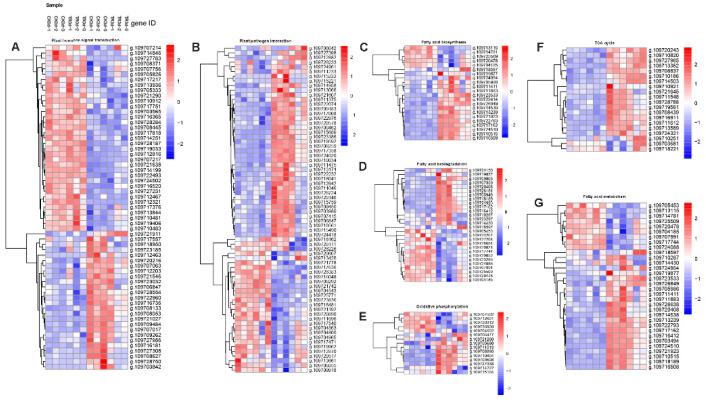
Heatmap of (**A**) plant hormone signal transduction, (**B**) plant–pathogen interaction, (**C**) fatty acid biosynthesis, (**D**) fatty acid biodegradation, (**E**) oxidative phosphorylation (**F**) TCA cycle (**G**) fatty acid metabolism.

**Figure 12 plants-11-02215-f012:**
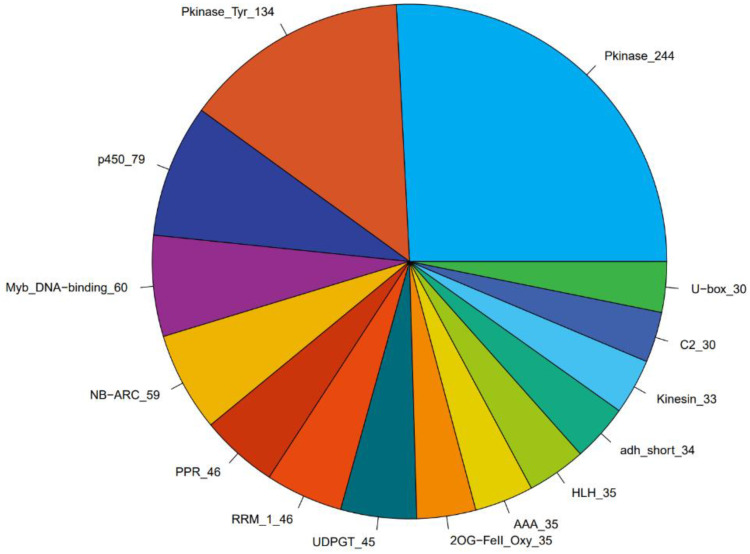
Proportion of transcription factors (15 gene families with the largest proportions).

**Figure 13 plants-11-02215-f013:**
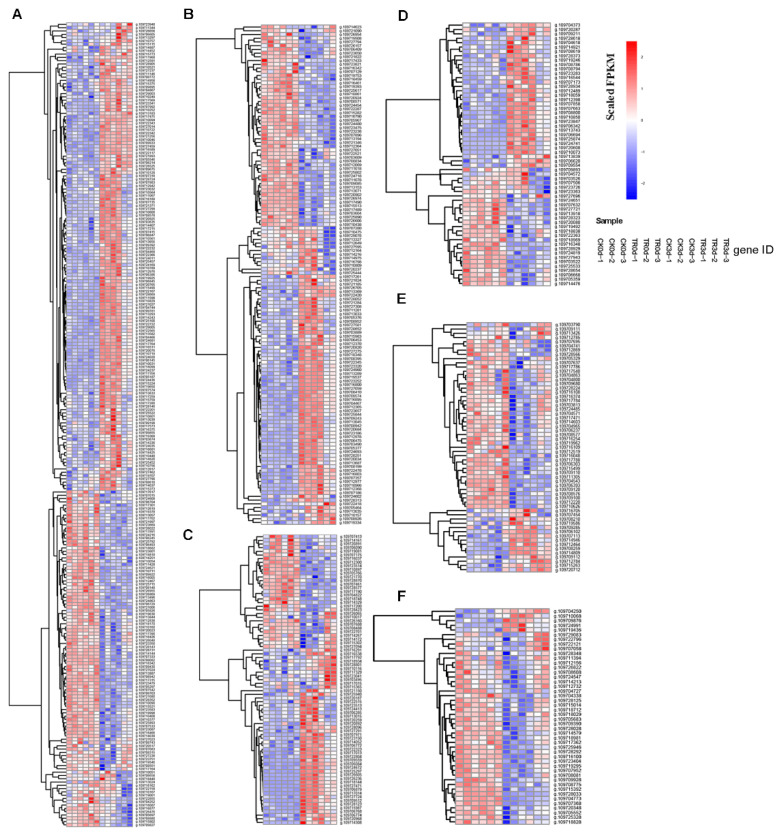
(**A**) Pkinase (244), (**B**)Pkinase Tyr (134), (**C**) P450 (79), (**D**) Myb DNA binding (60), (**E**) NB AR (59), (**F**) PPR (46). Heatmap of six gene families.

**Table 1 plants-11-02215-t001:** Transcriptome raw data for each sample, respectively, raw and clean reads, raw and clean bases, error rate, percentages of bases with Phred values greater than 20 in total bases, percentages of bases with Phred values greater than 30 in total bases, percentages of G and C in four bases in clean reads.

Sample	Library	Raw Reads (bp)	Raw Bases(G)	Clean Reads (bp)	Clean Bases (G)	Er%	Q20%	Q30%	GC%
CK0d-1	FRAS220060494-1r	47,917,660	7.19 G	46,984,726	7.05 G	0.03	96.74	91.51	49.72
CK0d-2	FRAS220052691-2r	41,150,130	6.17 G	40,322,240	6.05 G	0.03	96.54	91.25	49.05
CK0d-3	FRAS220052692-2r	42,604,436	6.39 G	41,244,352	6.19 G	0.03	96.87	91.82	48.73
TR0d-1	FRAS220052693-2r	45,285,588	6.79 G	44,454,528	6.67 G	0.03	96.74	91.55	48.64
TR0d-2	FRAS220052694-2r	45,354,954	6.8 G	44,426,710	6.66 G	0.03	97.03	92.09	48.68
TR0d-3	FRAS220052695-2r	46,247,152	6.94 G	45,288,912	6.79 G	0.03	96.95	91.96	47.58
CK3d-1	FRAS220052696-2r	45,892,678	6.88 G	45,465,842	6.82 G	0.03	96.99	92.16	50.89
CK3d-2	FRAS220052697-2r	46,614,824	6.99 G	45,989,324	6.9 G	0.03	96.8	91.62	49.54
CK3d-3	FRAS220052698-2r	45,824,306	6.87 G	44,867,480	6.73 G	0.03	96.77	91.59	49.78
TR3d-1	FRAS220052699-2r	55,346,256	8.3 G	52,010,944	7.8 G	0.03	96.85	91.85	50.09
TR3d-2	FRAS220052700-2r	41,487,432	6.22 G	37,793,338	5.67 G	0.03	97.09	92.33	49.92
TR3d-3	FRAS220060495-1r	45,300,832	6.8 G	44,175,794	6.63 G	0.03	96.4	90.89	51.06

## Data Availability

Data are available from the authors upon request.

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
