# Peer review of "Physiological and Transcriptome Analyses of CaCl2 Treatment to Alleviate Chilling Injury in Pineapple"

_plants, 2022, doi:10.3390/plants11172215_

Round 1

Reviewer 1 Report

The manuscript by Mengzhuo Zhang and co-autors titled “Physiological and transcriptome analysis CaCl2 treatment alleviates chilling injury of pineapple” presents the results of study carried out to understand the molecular mechanisms that induce pineapple chilling. The research is very interesting, methods of the experiment are completely and accurately described and statistical analyses are clear. In the Method chapter, complete the data on the number of fruits in each combination of the experiment. A statement such as "More than  ten fruits were sampled from each treatment after cold storage…" is inaccurate. The manuscript discussion is successful and the references are adequate. The experimental results are sufficient to justify the conclusions. In my opinion the paper is acceptable for publication in Plants.

Author Response

More than ten fruits were sampled from each treatment after cold storage ... is inaccurate.

Response: I also think scientific research needs to be rigorous. This is not accurate. I will change it to “Fifteen fruits were sampled from each treatment after cold storage 1 w transfer to room temperature 0 d,3 d and 6 d.There were three replicates to be included in each treatment.”.

Reviewer 2 Report

The thematic of the manuscript is very actual.

For improving of the quality there are some recommendation in the Methodical part.:

1. The quality of the fruits and their preservation is greatly influenced by the soil in which the plants were grown. It is recommended to provide a detailed description of what the soil was like, i.e. content of mobile phosphorus, mobile potassium, nitrogen, ph, humus content, etc. It is necessary to assess the meteorological conditions (temperature, rainfall, sunshine) during the vegetation period of the plants. Maybe the plantation was artificially watered? It is recommended to supplement this section in more detail.

2. Is it recommended to indicate how many fruits were selected for research?

Author Response

IThe quality of the fruits and their preservation is greatly influenced by the soil in which the plants were grown. It is recommended to provide a detailed description of what the soil was like, i.e. content of mobile phosphorus, mobile potassium, nitrogen, ph, humus content, etc. It is necessary to assess the meteorological conditions (temperature, rainfall, sunshine) during the vegetation period of the plants. Maybe the plantation was artificially watered? It is recommended to supplement this section in more detail.

Response 1:

For this suggest, I think this belong to fruits Pre-harvest problems. This problem also needs to be taken seriously, we purchase pineapple directly from Qionghai area, and there is no pineapple planted in the test area. Therefore, we can't realize the soil quality test. We are very sorry for this problem. However, we can provide the storage conditions for fruits preservation.

Storage condition:

Refrigerate: 6 ℃±1 ℃ Humidity:85 ± 5 % relative humidity (RH)

Room temperature: 25 ℃±1 ℃ Humidity:85 ± 5 % relative humidity (RH)

IIIs it recommended to indicate how many fruits were selected for research?

Response 2:

CK group: 150

50 CaCl2: 150

100 CaCl2: 150

150 CaCl2: 150

600 fruits were selected for research.

Reviewer 3 Report

In this interesting paper factor affecting the Intern Browning (IB) of pineapple after refrigerated storage during the shelf life causing chilling injury (CI) was investigated. In this study, pineapple IB related to ROS homeostasis, including organism relative balance between enzyme and non- enzyme antioxidants and oxides. Fruit quality, reactive oxygen species (ROS) and antioxidants and transcriptional performed to alleviating internal browning (IB) symptoms in pineapple fruit. Synthresis of phenilpropanoids, MAPK pathway, Plant hormone, Plant pathogen interaction, tricarboxylic acid cycle (TAC)
and Fatty acid biosynthesis are showed. good charts are presented  for instance GO and KEGG the up-regulated and down regulated genes in each comparison combination and Differential expression of oxidation and antioxidation related genes heatmap.

Comments please check English language, specially paragraphs are confusing

2.pictures of plants would be a plus

Figure 10 cannot be seeing please expand

Author Response

IComments please check English language, specially paragraphs are confusing.

Response 1:

Thank you for your precious comments,I will modify the language to make it clearer.

English language grammar problems have been rectified and is marked red in the manuscript.

IIPictures of plants would be a plusFigure 10 cannot be seeing please expand.

Response 2:

Sorry, you can't look over the Figure 10, I'll rearrange Figure 10 make it more distinct visible.